# Urbanization and cardiovascular health among Indigenous groups in Brazil

Anderson da Costa Armstrong [1,2 ✉], Carlos Dornels Freire de Souza[1,3], Juracy Marques dos Santos[2], Rodrigo Feliciano do Carmo [1], Dinani Matoso Fialho de Oliveira Armstrong[1], Vanessa Cardoso Pereira[2], Ana Marice Ladeia[4], Luis Claudio Lemos Correia[4,5], Manoel Barral-Netto [6,7] & Joao Augusto Costa Lima [5]

## Abstract

**Background** We described the prevalence of cardiovascular risk factors in groups of Brazilian Indigenous people at different degrees of urbanization.

**Methods** The Project of Atherosclerosis among Indigenous populations (Projeto de Atero-sclerose em Indígenas; PAI) is a cross-sectional study conducted in Northeast Brazil between August 2016–June 2017. It included three populations: Fulni-ô Indigenous people (lowest degree of urbanization), Truká Indigenous people (greater urbanization), and a highly urbanized non-Indigenous local cohort (control group). Participants were assessed to register sociodemographic, anthropometric, as well as clinical and laboratory-derived cardiovascular (CV) risk parameters. Age-adjusted prevalence of hypertension was also computed. Non-parametric tests were used for group comparisons.

**Results** Here we included 999 participants, with a predominance of females in all three groups (68.3% Control group, 65.0% Fulni-ô indigenous group, and 60.1% Truká indigenous group). Obesity was present in 45.6% of the urban non-Indigenous population, 37.7% Truká and in 27.6% Fulni-ô participants. The prevalence of hypertension was 29.1% ($n = 297$) with lower prevalence in the less urbanized Fulni-ô people (Fulni-ô – 18.2%; Truká – 33.9%; and Control – 33.8%; $p < 0.001$). In the elderly male population, the prevalence of hypertension was 18.7% in the Fulni-ô, 45.8% in the Truká, and 54.5% in the control group. Of the 342 participants that self-reported hypertension, 37.5% ($n = 68$) showed uncontrolled blood pressure (BP). Uncontrolled BP was more prevalent among Truká people when compared to Fulni-ô people and non-Indigenous participants (45.4%, 22.9%, and 40.7%, respectively; $p < 0.001$).

**Conclusions** We found a higher cardiovascular risk in communities with a higher degree of urbanization, suggesting that living in towns and cities may have a negative impact on these aspects of cardiovascular health.

### The plain language summary

The lifestyles and environments of traditional indigenous and city-living communities differ. We compared rates of obesity and hypertension in members of two under-studied Indigenous groups in Northeast Brazil and a nearby urbanized group. We found higher rates of obesity and hypertension amongst members of the more urbanized community, suggesting that living in towns and cities may have a negative impact on these aspects of cardiovascular health. These results suggest those living in the city should modify their lifestyle and monitor their cardiovascular health more carefully if possible.

[1] Universidade Federal do Vale do Sao Francisco, Petrolina, Pernambuco, Brazil. [2] Universidade do Estado da Bahia, Juazeiro, Bahia, Brazil. [3] Universidade Federal de Alagoas, Arapiraca, Alagoas, Brazil. [4] Escola Bahiana de Medicina e Saúde Pública, Salvador, Bahia, Brazil. [5] Johns Hopkins University, Baltimore, MD, USA. [6] Instituto Gonçalo Moniz - Fundação Oswaldo Cruz (Fiocruz), Salvador, Bahia, Brazil. [7] Faculdade de Medicina da Universidade Federal da Bahia, Salvador, Bahia, Brazil. ✉email: anderson.armstrong@univasf.edu.br

Cardiovascular diseases (CVD) are defined as a group of disorders of the heart and blood vessels—*coronary artery disease, heart failure, peripheral arterial disease, and cerebrovascular diseases*[1] and currently represent the leading causes of morbidity and mortality in Brazil[2] and worldwide[1]. They are deeply related to classic risk factors such as hypertension, obesity, smoking, and family history, besides other determinants such as socioeconomic status, ethnicity, diet, and lifestyle habits[1,2].

Traditional Indigenous communities are thought to be particularly vulnerable to the epidemiological transition from infectious diseases to chronic degenerative pathologies, including CVD as the most significant cause of morbidity and mortality[3,4]. Although Indigenous peoples worldwide face poorer health and social outcomes when compared to non-Indigenous populations[5,6], the real CVD prevalence among Indigenous people is largely unknown as these race-ethnic groups are systematically underrepresented in population studies.

CVD mortality continues to rise among Indigenous populations living in the São Francisco River basin, Brazil, while in local urbanized cities, CVD mortality has trended towards stabilization over recent years[3]. Globally, less urbanized groups appear to be less susceptible to CVD when compared to similar ethnic groups living in highly urbanized areas located in the same geographic regions[1,7].

Indigenous populations are exposed to a greater degree of pragmatic social vulnerability. There are known continuing effects of territorial and cultural loss that affect Indigenous peoples' health[8]. It is, however, still largely unknown how urbanization relates to risk factors in these vulnerable populations who usually have important cultural and social dynamics[3].

In the PAI (Project of Atherosclerosis among Indigenous Groups) study, we hypothesized that the prevalence of CVD risk factors would reflect the evolving differences related to urbanization among the diverse Indigenous groups still living in the São Francisco River basin. Therefore, we assessed the prevalence of CVD risk factors in different Indigenous and non-Indigenous communities co-located in the basin but exposed to different degrees of urbanization. In this study, we described the prevalence of cardiovascular risk factors in groups of Brazilian Indigenous people at different degrees of urbanization. In summary, our results suggest that communities exposed to a higher degree of urbanization have a higher cardiovascular risk.

## Methods

**Study design and recruiting.** The Project of Atherosclerosis among Indigenous populations (PAI) is a cross-sectional study conducted in Northeast Brazil between August 2016 and June 2017. The study included 999 people living in three communities of markedly different degrees of urbanization in the São Francisco River basin (Fig. 1): the lowest degree of urbanization found in the Indigenous Fulni-ô People ($n = 303$; 30.3%); the intermediate Truká Indigenous group ($n = 336$; 33.7%); as well as a highly urbanized cohort living in the city of Juazeiro ($n = 360$; 36.0%) located at the margin of the São Francisco River. The study integrated established cultural and geographical parameters used to characterize and measure the magnitude of urbanization[9].

The PAI study protocol has been detailed previously[10]. Briefly, we included women and men aged between 30 and 70 years old, residents of the Indigenous Fulni-ô and Truká communities or in the city of Juazeiro. We excluded those with clinically manifested heart failure, history of coronary or cerebrovascular vascular diseases requiring hospitalization, renal failure on dialysis, history of surgery for peripheral arterial disease or heart disease, and participants who expressed reluctance to undergo diagnostic tests including blood draw. As the investigation seeks to identify risk factors and subclinical diseases, the adoption of these criteria was necessary.

The PAI study was approved by the National Research Ethics Council (CONEP number 1.488.268), the National Indigenous Foundation (Fundação Nacional do Índio [FUNAI]; process number 08620.028965/2015-66), and the Indigenous leaders of both participating groups. All participants provided written informed consent before enrollment in the study.

*Population groups.* The reference urbanization (fully urbanized population) was defined as the population residing in the city of Juazeiro, Bahia, while the two indigenous populations (Fulni-ô and Truká) occupy rural areas. The classification of the degree of urbanization adopted in this study was based on the characteristics of the group, such as: a. geographical location; b. Proximity and contact with cities; c. maintenance of traditional culture; and d. influence of the city on the group's dynamics[9].

The Fulni-ô People, least urbanized group, live in an Indigenous reserve occupying 11,505 hectares at the margin of the Ipanema River, a tributary of the São Francisco River, located in the semi-arid region in Northeast Brazil[11]. After repeated waves of outward migration over several decades, part of the original group remained in this territory and organized communities to hold on to their traditions[11,12]. The Fulni-ô group is characterized by its geographical location, with access by side roads, reduced contact with the population of nearby cities, maintenance of traditional culture, and reduced influence of the city on the social dynamics of the group[11,13]. They are the only Indigenous group in Northeast Brazil still using their original traditional language (Yate) for daily communication, although they also speak Brazilian Portuguese. Moreover, the Fulni-ô people still spend 3 months of every year in isolation for the Ouricuri traditional ritual. During this time, they refrain from all interactions with non-Indigenous individuals[13].

The Truká people participants, the group with an intermediate degree of urbanization, were recruited in Assunção Island, the largest island in the São Francisco River[14,15]. They have been reported to live in this territory since the 18th century and obtained rights for a specific reservation area in 2002. The Truká People territory has witnessed major local infrastructural interventions including the construction of dams and large canals over several years with significant agricultural changes. Truká people participants still use fragments of their traditional language during rituals, but regular Brazilian Portuguese in daily routine communication[16]. All indigenous people living in the Fulni-ô and Truká communities were invited to participate in the PAI study in advance, using community meetings, indigenous leadership communications, and the available media.

The urban non-indigenous population was recruited in the city of Juazeiro, Bahia. Juazeiro is a municipality in the state of Bahia, located in the São Francisco Valley Mesoregion. It is the sixth most populous municipality in Bahia and the tenth in the interior of the Northeast. The city stands out for its irrigated agriculture, being one of the main producing centers of tropical fruits in the country[17,18]. In 2020, the population was estimated in 218,162 people with a population density of 30.45 people/km$^2$; the urbanization level is 81.21%, with 64.2% of houses with appropriate sanitation[18].

The characteristic populational mixing that has happened in Brazil since European colonization has included the migration of Indigenous peoples to urbanized communities, resulting in similar anthropometric phenotypes when comparing local Indigenous individuals with the non-Indigenous population of Juazeiro city. However, Indigenous groups differ from non-Indigenous ones as they are firmly bound to their territories, beliefs, and indigenous traditions[18].

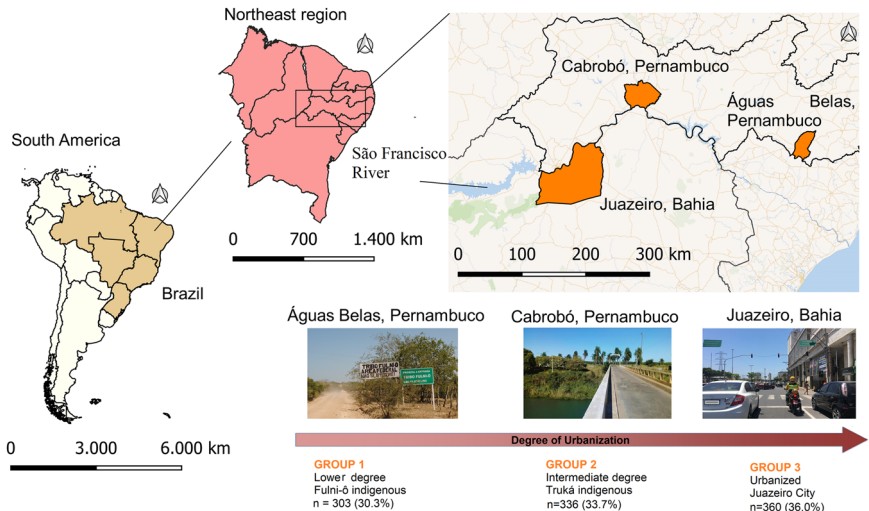

**Fig. 1 Study population and territorial area, for Fulni-ô Indigenous people, Truká Indigenous people, and urban non-Indigenous population in Juazeiro.** Made with Natural Earth. Map base layers were modified in QGIS software version 2.18. Pictures from the PAI researchers (no Indigenous person shown).

When initial study planning was conducted, information regarding Juazeiro City districts was directly assessed in official public records available in the city hall. Recruiting was conducted in central neighborhoods of Juazeiro (Centro e Angari), with an estimate total of 3270 inhabitants (similar to the 2570 inhabitants in the assessed Truka community and to the 3254 Fulnio indigenous people). We selected two historical neighborhoods in the Juazeiro city, with lifelong residents. These areas are considered symbolic for the city growth, as most of the city expansion initiated from these central neighborhoods. Thus, these neighborhoods are the most representative of the original inhabitants of Juazeiro.

Importantly, in all three groups (Fulni-ô, Truká, and Juazeiro peoples), convenience sampling included men and women between 30 and 70 years old, to assess a more homogeneous cardiovascular risk base and to avoid age-related bias. Recruitment took place through invitations made to residents through neighborhood associations and local leaders, as well as radio and television networks. In the case of indigenous populations, prior contact was made with local leaders and visits were scheduled. None of the people that voluntarily presented to the researchers refused to participate. All individuals who agreed to participate in the research provided a signed consent form.

*Sociodemographic and anthropometric parameters.* We registered sex as a binary variable (male/female). Age was computed as a continuous variable in years.

Individuals were classified according to BMI (Kg/m$^2$) as underweight (<18.5), normal (≥18.5 and <25), overweight (≥25 and <30), and obese (≥30)[19]. Then, the prevalence of obesity was stratified by sex and age group (30–39 y/o; 40–49 y/o; 50–59 y/o; and 60–70 y/o). Body fat was also assessed by computing neck circumference (NC), waist circumference (WC), and waist-to-hip ratio (WHR). Abnormal values for NC were ≥37 cm for men and ≥34 cm for women; for WC were ≥94 cm for men and ≥80 cm for women; and for WHR were ≥0.95 for men and ≥0.85 for women[20–22].

*Cardiovascular risk assessment.* Individuals were interviewed regarding the self-reported diagnosis of hypertension, as well as regarding the use of medications. Hypertension diagnosis was established if blood pressure were equal or higher to 140 × 90 mmHg[23], as measured by the researchers or if the participant also referred the use of blood pressure medication. Mean arterial pressure (MAP) was obtained indirectly, through the following equation: MAP = DBP + 1/3 (SBP−DBP), where:

MAP = Mean Blood Pressure, SBP = Systolic Blood Pressure and DBP = Diastolic Blood Pressure[24]. Then, hypertension prevalence was stratified by sex and age group (30–39 y/o; 40–49 y/o; 50–59 y/o; and 60–70 y/o). We also intended to investigate whether those participants that were aware of their CV risk factor would be adequately controlled. For this, those who auto-referred a known diagnosis of hypertension were assessed and classified as under control if BP < 140 × 90 mmHg.

Peripheral arterial disease was evaluated using ankle-brachial index (ABI). The ABI values were computed as previous recommendations and categorized as normal (≥0.9 and <1.30) or abnormal (<0.90 and/or >1.30)[23]. Abnormal ABIs were later graded in mild peripheral arterial disease (PAD; when between 0.71 and 0.9); moderate PAD (0.41–0.7); severe PAD (≤0.4); and arterial calcification when ≥1.4[23].

*Statistics.* Continuous quantitative variables were presented through central tendency and dispersion (mean ± standard deviation; median and interquartile range—IQR; minimum and maximum) and qualitative through frequencies (absolute and relative). The number of participants with available data is shown in the tables and Figures. The following statistical tests were used: Shapiro-Wilk for data distribution evaluation; *U*-Mann-Whitney for comparison of quantitative variables between two groups; Kruskal-Wallis test for comparing quantitative variables between three groups (and post-hoc, when necessary); and chi-square test with continuity correction for categorical variables. The prevalence of age- and sex-adjusted hypertension was computed using direct standardization[25], with an internal method considering the Brazilian population, to reduce the influence of those variables in the prevalence of hypertension. Confidence intervals of 95% and a significance level of 5% were used in the analyses. Significant associations were considered when *p* < 0.05.

**Reporting summary.** Further information on research design is available in the Nature Portfolio Reporting Summary linked to this article.

## Results

A female predominance was observed in the three groups (65.0% in the Fulni-ô, 60.1% in the Truká and 68.3% in the Control group). The Fulni-ô indigenous participants were older than the non-indigenous (*p* = 0.008) and the Truká participants (*p* = 0.033). No

**Table 1 Characterization of the studied groups.**

| Variable | Fulni-ô | Truká | Control group | P value |
|---|---|---|---|---|
| Degree of urbanization | Low | Intermediate | High | |
| Age (median; IQR) | 46.0; 10.8 | 48.0;16.0 | 49.3; 15.0 | 0.045[a]* |
| Sex | | | | 0.076[b] |
| Female | 197 (65.0%) | 202 (60.1%) | 246 (68.3%) | |
| Male | 106 (35.0) | 106 (39.9%) | 114 (31.7%) | |
| BMI (median; IQR) | 27.3; 6.0 | 28.2;5.4 | 29.2;6.9 | <0.001[a]* |
| BMI groups | | | | |
| Underweight | 1 (0.3%) | 0 (0.0%) | 1 (0.2%) | - |
| Normal | 84 (27.7%) | 84 (25.0%) | 73 (20.2%) | |
| Overweight | 131 (43.3%) | 127 (37.8%) | 122 (33.9%) | |
| Obesity (I, II, and III) | 87 (28.7%) | 125 (37.2%) | 164 (45.6%) | |
| (n = 933) | | | | |
| NC (cm) (median; IQR) | (36.7; 5) | (37; 5) | (35; 5) | 0.001[a]* |
| Abnormal NC – n (%) | 215 (74.6%) | 232 (73.4%) | 226 (68.6%) | 0.212[b] |
| WC (cm) (median; IQR) | (91; 14.0) | (96; 16) | (95; 15) | <0.001[a]* |
| Abnormal WC – n (%) | 175 (67.3%) | 261 (78.8%) | 285 (82.3%) | <0.001[b]* |
| HC (cm) (median; IQR) | (100; 13) | (102; 14) | (101; 13) | 0.382[a] |
| Abnormal WHC – n (%) | 131 (52.8%) | 183 (58.4%) | 212 (66.8%) | 0.003[b]* |
| n = 944 | | | | |
| SBP (median; IQR) | (125.5; 23) | (134; 28) | (133; 25) | <0.001[a]* |
| DBP (median; IQR) | (76.0; 13.0) | (80; 16.5) | (80; 13.5) | <0.001[a]* |
| MAP (median; IQR) | (92.7; 16.6) | (98.7; 19.4) | (97.7; 19.4) | <0.001[a]* |
| n = 908 | | | | |
| ABI (median; IQR) | (1.01; 0.1) | (1.12; 0.1) | (1.08; 0.1) | <0.001[a]* |
| Abnormal ABI – n (%) | 14 (5.4%) | 18 (5.4%) | 24 (7.4%) | 0.490[b] |

*BMI* body mass index, *NC* neck circumference, *WC* waist circumference, *HC* hip circumference, *WHR* waist-to-hip ratio, *SBP* systolic blood pressure, *DBP* diastolic blood pressure, *MAP* mean arterial pressure, *ABI* Ankle-brachial index.
[a]Kruskal-Wallis test.
[b]chi-squared continuity correction.
*$p < 0.05$.

age difference was found between Truká and non-Indigenous participants (Table 1).

All three groups were on average overweight (BMI ≥ 25 and <30 kg/m$^2$). The urban non-Indigenous population had the highest BMI, and the less urbanized Fulni-ô had the lowest (median 29.2; IQR 6.9 kg/m$^2$ vs. 27.3; IQR 6.0 kg/m$^2$, respectively; $P < 0.001$). Normal BMI was found in 24.1% ($n = 241$) of the participants, with a higher frequency for the Fulni-ô people (27.7%) when compared to the Truká people (25.0%) and the urban non-Indigenous population (20.2%) (Table 1). When BMI was stratified according to age groups, the control group showed significant differences when comparing the youngest with the oldest (30–39 y/o [30.4 ± 5.8 kg/m$^2$] vs 60–70 y/o [27.6 ± 5.0 Kg/m$^2$]; $p = 0.024$).

The prevalence of obesity was 45.6% in the urban non-Indigenous population, 37.2% in Truká and 28.7% in Fulni-ô participants (Fig. 2). The prevalence of obesity was higher in women when compared to men in all three groups. Still, it differed among the studied groups in both sexes, increasing with urbanization (Supplementary Data 1).

Non-indigenous participants had more WHR abnormal values, when compared to indigenous peoples ($P = 0.003$; Table 1). In men, neck circumference (NC) was higher among the Fulni-ô Indigenous compared to urban non-Indigenous population ($P = 0.038$), and waist-hip ratio (WHR) was higher among the Truká Indigenous people compared to the other groups. In women, waist circumference (WC) ($P = 0.001$) and WHR were greater in non-Indigenous urbanized population (Supplementary Data 1).

Peripheral arterial disease was evaluated in 904 individuals (90.49%): 254 (28.1%) Fulni-ô, 328 (36.3%) Truká, and 322 (35.6%) non-Indigenous population. In all three groups, mean ABI values were classified as normal. Abnormal ABI was found in

6.2% ($n = 56/904$) participants, being 66.1% ($n = 37$) females. Of these, 42.8% ($n = 24$) were in the urban non-Indigenous population. Additionally, there were no differences in the prevalence of altered ABI in women versus men or among the three study groups (Supplementary Data 1).

The prevalence of hypertension as assessed by high blood pressure measures and/or use of medication was 29.1% ($n = 297$), with lower prevalence in the less urbanized Fulni-ô people (Fulni-ô –18.2%; Truká – 33.9%; and Control – 33.8%; p < 0.001; Fig. 2). The highest prevalence of age- and sex-adjusted hypertension was found among non-indigenous urbanized participants (32.0%) and the lowest was found for the Fulni-ô indigenous participants (17.0%; Fig. 2). When stratified according to sex and age group, the prevalence of hypertension increased with age in both sexes, being lower in the Fulni-ô population. In the elderly male population, the prevalence of hypertension was 18.7% in the Fulni-ô, 45.8% in the Truká, and 54.5% in the control group. In the elderly female population, the prevalence was higher in the Truká when compared to Fulni-ô (47.4% vs. 42.2%, respectively; $p < 0.001$).

The prevalence of self-reported hypertension was observed in 34.2% ($n = 342$) of the total population. Of these, 37.5% ($n = 68$) had uncontrolled blood pressure. Uncontrolled BP was more prevalent among Truká people when compared to Fulni-ô people and non-indigenous participants (45.4%, 22.9%, and 40.7%, respectively; $p < 0.001$).

## Discussion
This study pioneers the investigation of cardiovascular diseases in a large number of Northeast Brazil Indigenous and non-Indigenous people living in diverse degrees of urbanization. Importantly, we showed that different Indigenous groups in geographical proximity

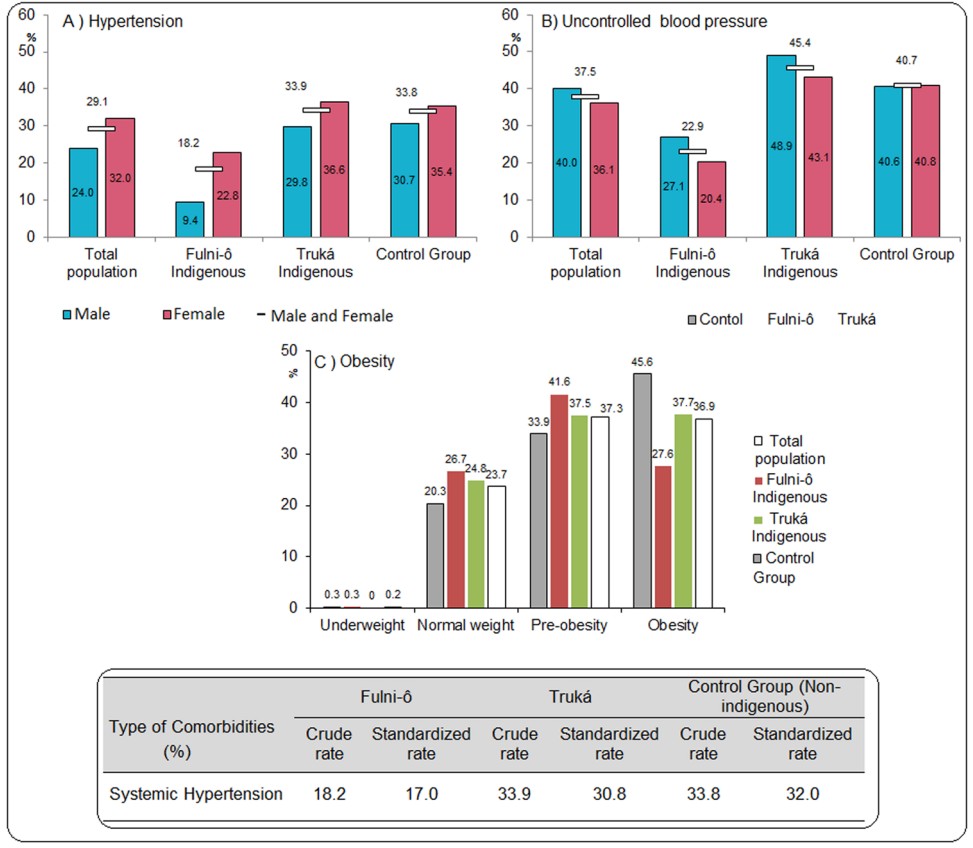

**Fig. 2 Prevalence of hypertension and obesity diseases for Fulni-ô Indigenous people, Truká Indigenous people, and urban non-Indigenous population.** Brazil. **a** Hypertension; **b** Uncontrolled blood pressure; **c** obesity. Total population ($n = 999$): the least urbanized Fulni-ô People ($n = 303$; 30.3%); the intermediate urbanized Truká Indigenous group ($n = 336$; 33.7%); as well as a highly urbanized cohort living in the city of Juazeiro ($n = 360$; 36.0%). *Control group refers to the urban non-Indigenous population.

are exposed to a diverse cardiovascular risk, partly explained by the degree of urbanization.

The history of Brazilian colonization and the recent wave of urbanization resulted in territorial losses to indigenous peoples and promoted important changes in the social structure and in habits and customs[26]. In most indigenous communities, urbanization led to changes in economy, social organization, and habits[3,26,27]. The Northeast coast was the primary area to undergo European colonization in Brazil during the 16th century, using the São Francisco River to access inner territories[11,16]. The Indigenous communities that live by this river have been gradually affected by increasing contact with non-Indigenous populations and major infrastructural interventions[3,9,28].

In general, we found that the less urbanized Fulni-ô people had the most favorable CV risk profile. The pre-colonial Fulni-ô people used to occupy a vast territory nearby the São Francisco River[11,12]. The Fulni-ô have been driven away from their original land. They now live on the banks of a minor affluent, the Ipanema River. Such geographical move might have aided this community to maintain a less urbanized way of living, as they fled from the areas with intense urbanization processes. The Fulni-ô people tend to maintain a diet based on natural and wholesome food and rural work as part of their livelihood[11,12].

Obesity was a CV risk factor that increased according to urbanization, particularly among Indigenous women[12]. Remarkably, the average BMI in older Indigenous people was lower than in younger groups. Such an unexpected observation might be related to an ongoing generational change in lifestyle and may reflect a nutritional transition process, with greater exposure of the younger population to nutritional changes. The reduced

number of underweight individuals ($n = 2$) indicates that phase 1 of the nutritional transition may have already been overcome in the groups studied. This fact, added to the higher BMI in women, reinforces the fact that the groups are entering phase 2 of the nutritional transition. The Indigenous groups tend to get more westernized along the years of rapid urbanization. As the exposure to risk factors tends to evolve, future studies may find a smaller gap in CV risk when comparing Fulni-ô and Truká people.

Urbanization has been previously related to increases in obesity for the general Brazilian population, with predominance in women. In 2013, the National Health Survey in Brazil showed that the prevalence of obesity in urban areas reached 13.2% for men and 17.0% for women, higher than the prevalence of 8.8% in men and 16.5% in women observed in rural areas[29]. In addition, it is possible that the predominance of recruited women in all groups may be explained by the fact that men are usually less likely to pursue health care[7,12,15].

The role of raising children in indigenous societies is typically attributed to women and body fat increases are strongly associated with changes in nutritional profiles that emerge early in human lives. Thus, our findings of a higher presence of obesity in indigenous women may magnify the problem in future generations. Importantly, even though the Fulni-ô group had a better cardiovascular profile, the prevalence of obesity observed was higher than the obesity reported in these highlighted studies.

If, on the one hand, the exposure to CVD risk factors is more intense in urbanized areas, on the other hand, it ensures greater access to health services. Thus, greater contact with cities enables the early diagnosis of diseases. These differences between urban

and rural areas also reflect the access to health services by the populations[30]. In the case of indigenous populations, this problem is in part mitigated by the existence of a specific subsystem for basic health care. However, the low number of professionals and the lack of specialized health care are still reported as major issues for Northeast Brazil indigenous populations[31,32].

We found a high prevalence of hypertension in the studied Indigenous groups. Although the literature on hypertension among Brazilian Indigenous people is scarce, a systematic review from 1970 to 2014 found a 12% annual relative increment in the probability of a Brazilian Indigenous person developing hypertension[8]. Importantly, the overall prevalence of hypertension was reported in 6.2%, much lower than our findings (18.2% for Fulni-ô and 33.9% for Truká).

The vast majority of the Truká people aware of their previous diagnosis of hypertension was not with controlled blood pressure levels. The Indigenous Truká people had a similar prevalence of hypertension to the non-Indigenous people, but the Indigenous people showed a much lower success in blood pressure control. It might be explained by the fact that Truká people are already exposed to a more westernized lifestyle, similar to the nearby highly urbanized city. However, the Truká people lack access to adequate health care and health education.

Many countries have adopted measures that aim to mitigate the harmful effects of the urbanization process on indigenous health. Australia, Canada and New Zealand reinforce health education programs at the level of primary health care[33,34]. In Brazil, the National Policy for Indigenous Health was launched in 1999—a health subsystem organized in Special Indigenous Health Districts (Distritos Sanitários Especiais Indígenas – DSEI, in Portuguese), under the responsibility of the Special Secretariat for Indigenous Health (Secretaria Especial de Saúde Indígena – SESAI, in Portuguese)[35]. This subsystem, articulated with all other components of the Unified Health System (Sistema Único de Saúde – SUS, in Portuguese) is responsible for providing health care to the country's indigenous populations.

This study has relevant limitations. The study design (prevalence) does not allow the identification of estimates of the CV risk in the groups. Recruiting participants in indigenous settings is challenging, as it relies on voluntary participation and it also has potential selection bias. A higher number of female participants might have impacted the cardiovascular risk burden, as men tend to have more risk factors and events in the studied age range. There are also intrinsic limitations related to a cross-sectional study, unable to address causality in the findings.

In conclusion, the less urbanized Fulni-ô Indigenous people had a more favorable CV risk profile when compared to intermediate urbanized Truká and highly urbanized non-Indigenous peoples. Our findings indicate that CV risk is worse in Indigenous groups as the degree of urbanization is higher. Public health policies toward health care and health education are needed to address urbanization-related cardiovascular disease among Indigenous populations.

There are still large gaps in knowledge about indigenous health in Brazil. However, this study, by showing the scenario from a broad perspective, can contribute to the strengthening of public policies for health care and promotion that allow mitigating the damage of the urbanization process in the preservation of indigenous health, respecting their historic and cultural backgrounds.

## Data availability

Data on Brazilian indigenous peoples are restricted by several regulations. Therefore, we are not allowed to freely distribute our dataset. Any researcher interested in accessing our dataset must seek authorization from the official regulatory agency: Fundação Nacional

do Índio – Funai (https://www.gov.br/funai/pt-br). The source data for Fig. 2 is available in Supplementary Data 2.

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

## Acknowledgements

We acknowledge the collaboration efforts from the Indigenous peoples recruited in this study. We acknowledge the work of the PAI students and staff. We acknowledge the support from CARDIOVASF – Instituto do Coração do Vale do São Francisco. The study was funded by two Universal grants (2014/2018) and the Science Without Borders Program, all from Conselho Nacional de Desenvolvimento Científico e Tecnológico-CNPq- Ministry of Science, Technology, Innovations and Communications of Brazil. The study was also supported by Fundação Maria Emilia (Salvador, Bahia Brazil) and Universidade Federal do Vale do Sao Francisco (Petrolina, Pernambuco, Brazil). The funders had no role in study design, data collection and analysis, decision to publish, or preparation of the manuscript. M.B-N. is a research fellow from CNPq.

## Author contributions

A.C.A., J.A.C.L., M.B-N., D.M.F.O.A. conceived the idea for the study. All authors contributed to the study design. M.B-N., A.A., and A.M.L. secured funding for the study. A.C.A., D.M.F.O.A., V.C.P., R.F.C. acquired study data. A.A., D.M.F.O.A., C.D.F.S., J.S., L.C. drafted the manuscript. All authors critically revised the manuscript and approved the final version for submission.

## Competing interests

The authors report no potential competing interest.
