## [Peer Review File · Communications Medicine]

Reviewers' comments:

Reviewer #1 (Remarks to the Author):

The study describes the prevalence of cardiovascular risk factors in two Brazilian Indigenous population and a non-Indigenous population, which have different degrees of urbanisation. The authors found that less urbanised Fulni-o Indigenous had a more favourable cardiovascular risk profile when compared to intermediately urbanised Indigenous Truka and highly urbanised non-Indigenous population.

The topic is relevant, especially given the lack of literature in Brazilian Indigenous populations, and the data used has great value. However, I have some suggestions to improve the quality of the manuscript.

Major comments:

- The study included 999 participants, from which 303 (30.3%) were Indigenous Fulni-o, 336 (33.7%) were Indigenous Truka, and 360 (36.0%) were non-Indigenous. Although details on the study have been described elsewhere, it is important to mention briefly. How was the selection process? Were there any refusals, missing? A flow chart of participation would be helpful.
- The groups assessed have different degrees of urbanisation; one group is defined as the 'least urbanised group', other as 'intermediate degree of urbanisation', and the other as 'highly urbanised'. What is the proportion of urbanisation in each of those groups? This contextual information is important.
- Body mass index is classified into 6 groups, but the sample is relatively small and some groups have a small number of participants; I suggest using only 3 categories (normal, overweight and obesity)
- Biomarkers were assessed in only a small part of the participants (varying from 6% in Truka men to 22.8% in Fulni-o women). It is unlikely that missing is at random, and those with information on biomarkers might differ from their original groups in several characteristics. Information regarding missing should be described.
- Similarly, it seems that only 56% of the population had information on smoking, and this was only 24% in the non-Indigenous population. Why is there so much missing? Again, a flow chart would be helpful. The authors should reconsider the use of smoking and biomarkers information, given this is likely affected by selection bias.
- Why non-parametric tests were used? Furthermore, several comparisons have been performed, which increases type 1 error. I suggest to use parametric tests and compare estimates with a single reference group (non-Indigenous).
- The results are confusing and seem to deviate from the objective of the manuscript, which was to describe the prevalence of cardiovascular risk factors in the different groups. The prevalence of several risk factors is also compared by sex within each group and sometimes between groups, and also by age groups (with pairwise comparisons). However, in addition to differing by degrees of urbanisation, the groups also differ in relation to sex and age. Therefore, age- and sex-standardised cardiovascular risk factors should be presented.
- The figures presented are not very informative. Results from figures 2-5 can be combined in a table.
- "Cardiometabolic disease was observed in 40.4% of the total population". How was cardiometabolic disease defined?

- What is SAH (Figure 5)?
- It is surprising that 90% of the non-Indigenous population has high total cholesterol, and this prevalence seems to be <50% in the Brazilian population (PMID: 28658385). However, only 9.4% of the non-Indigenous population had information for total cholesterol. It is difficult to believe that these results (not only cholesterol, but also the other biomarkers) have not been affected by selection bias.

Minor comments:

- The authors mention that Ingenious Fulni-o are the only Indigenous group who still use their original traditional language. Was the questionnaire applied in their original language?
- What is mean blood pressure (MBP)? Mean arterial pressure (MAP)? How was it calculated?

Reviewer #2 (Remarks to the Author):

1. Abstract:

The proposed investigation of CV risk burden determinants was not object of the investigation, since no data were presented about mortality or disability. The abbreviation of kilo is kg, with lower case.

2. Methods:

2.1- The sample size needs more details. What was the total population of each group ? What fraction of the population represents each study group ? How was determined the sample size, in particular of the control group? How was the the refusal rate?

What was the reason to exclude those with clinically heart failure, coronary heart disease or cerebrovascular disease ? If the purpose is a cohort study it seems reasonable to have population free of the disease at the beginning, but not in a prevalence/cross-sectional study.

2.2- Laboratory testing. The normal values for HDL-c for non-fasting bood samples needs revision. The method for HbA1c needs to be informed since there is a greatt variability in the quality of this exam.

2.3- There is no information about the estimation of burden of CV risk factors.

3. Results

3.1- There is no reason to present average results for the sum of the three groups, since they are distinct groups.

3.2- Since the groups have distinct age-structure, prevalence rates should be presented as age-adjusted rates, or by age-groups.

3.3- p-values should be presented with p in lower case.

3.4- When presenting prevalence rates of cardiometabolic diseases, it should be considered the cases that were excluded because they had these kind of diseases. The way was done induces some underestimation.

3.5- No data for burden estimation, as mortality and disability, were presented

4. Discussion

4.1- In spite of sparse, there are several publications about hypertension in Brazilian Indigenous population after 2014.

4.2- Data presented did not allow to state that "CV risk burden is worse in Indigenous groups as the degree of urbanization is higher".

4.3- Among limitations of the study, it should be considered the sampling procedure used.

5. Figures and Tables

5.1- Figures are in excess and should not present results for the sum of the three groups.

5.2- Table 1 = categories of obesity should be grouped as obesity, since the number of individuals are fewer in some strata,

6. References :

6.1- Need some up date

6.2- Reference 2 should have name off countries in capital letters.

Reviewer #3 (Remarks to the Author):

Hi there,

I read your paper with interest. This paper looked at urbanisation and cardiovascular health among very interesting cohorts in Brazil. The paper is generally well written. The study design is appropriate for this research.

I have the following suggestions,

1. It is not clear how urbanisation is defined and measured in this study. A clear definition or discussion would be useful in this context.

2. To make it appealing to a wider audience, It would be interesting to discuss the findings here in relation to other similar studies looking at other developing countries, as similar findings were documented in other countries.

3. Urbanisation may also bring positives that may benefit health outcomes, through new job, non-manual, opportunities and better infrastructure. So there might be a positive force associated with urbanisation on socioeconomic factors were neglected in the discussion. I would like to see some of these literature in the discussions.

Regards

Reviewer #4 (Remarks to the Author):

The current manuscript entitled “Urbanization and cardiovascular health among Indigenous groups in Brazil: The Project of Atherosclerosis among Indigenous populations (PAI)” aims to describe the prevalence of cardiovascular risk factors in groups of Brazilian Indigenous individuals at different stages of urbanization and investigate these groups’ CV risk burden. Although the topic is of great interest because it addresses Indigenous populations in Brazil, who are not contemplated in most epidemiological studies and whose epidemiological transition may be delayed according to urbanization, the burden of infectious disease, nutrition and access to health, it requires clarifications, and would also benefit from English editing. The authors concludes that the less urbanized Fulni-O ethnic group has a more favorable CV risk profile, when compared to the intermediately urbanized and the highly urbanized groups.

MAJOR

1. Although the findings are very interesting and corroborates the concept that urbanized lifestyles lead to loss of CV health – which is particularly relevant to indigenous individuals –, the effect of age should be better explored. It is expected that cardiometabolic risk factors increase with age and the age structure of the populations seem to differ. Fulni-O might be healthier just because they are younger, or it might be interesting to depict the later epidemiological transition if the younger have a worse profile than the older individuals, particularly if the age effect is different across populations. As such, I would like the authors to comment on two aspects: 1) for the comparison across groups, a better approach would be to adjust the findings regarding BMI, cholesterol, hypertension and diabetes for age; 2) It is interesting that younger individuals had higher BMI than their older counterparts. This could be a result of a more recent nutritional transition, but may also be a selection bias or a survival effect bias, when the older individuals who were obese might have died. More detailed implications of the different populations' age structure should be added to the manuscript. Women might have this worse profile also because they were older than men.

2. Although the authors argue that Fulni-O individuals had a better CV profile than the other comparison groups, still CV health is poor in the Fulni-O population, particularly when comparing to other data from cohorts and nationally representative surveys in Brazil. In the last VIGITEL study, the prevalence of self-reported obesity was 17%, while in the ELSA-Brasil cohort studies, which included individuals in 6 Brazilian capital cities, the prevalence of measured obesity was 22% (age range 35-74y, mean age 50y, 51% female). Even in the light of the known heterogeneity across the individuals included in these studies and the PAI study, it is important to highlight that the prevalence of obesity was considerably higher than the reported in other studies even for the Fulni-O population, despite a younger population. The same is true for abdominal obesity. This could be a result of the inclusion criteria, as such how the selection of the individuals for participation in the study was made should be further explained and evaluated as a possible limitation.

3. Regarding BMI, two interesting points should be evaluated by the authors: 1) there were no individuals underweight, suggesting that the phase of malnutrition that usually occur in the first phase of epi transition has indeed been overcome; 2) Women had higher BMI than men: this is usually described in earlier stages of the second phase of nutritional transition – women go through the nutritional transition before men (maybe due to the fact that they have more sedentary roles). Another explanation may be due to selection bias, as the findings differ from what was reported in the PNS, VIGITEL and ELSA-Brasil. Please elaborate on that.

3. Could the relation of hypertension with salt consumption be further explored?

4. The losses for each analyses should be further detailed. It seems that only a minority of individuals had lab results, however this is not clear and it is a potential limitation that must be acknowledged.

REBUTTAL LETTER

Dear Editor,

We appreciate very much for the kind consideration of our manuscript for publication.

We thank the reviewers for their thorough work. We have tried to reply to the comments of the reviewers to our best as shown underneath and also included in the text.

We hope that our manuscript is now fit for publication.

Thank you in advance.

Reviewer #1 (Remarks to the Author):

The study describes the prevalence of cardiovascular risk factors in two Brazilian Indigenous population and a non-Indigenous population, which have different degrees of urbanisation. The authors found that less urbanised Fulni-o Indigenous had a more favourable cardiovascular risk profile when compared to intermediately urbanised Indigenous Truka and highly urbanised non-Indigenous population.

The topic is relevant, especially given the lack of literature in Brazilian Indigenous populations, and the data used has great value. However, I have some suggestions to improve the quality of the manuscript.

Major comments:

- The study included 999 participants, from which 303 (30.3%) were Indigenous Fulni-o, 336 (33.7%) were Indigenous Truka, and 360 (36.0%) were non-Indigenous. Although details on the study have been described elsewhere, it is important to mention briefly. How was the selection process? Were there any refusals, missing? A flow chart of participation would be helpful.

A: Thanks for the comment. We have added a paragraph explaining the selection process in Methods (Page 5): “All indigenous people living in the Fulni-ô an Truká communities were invited to participate in the PAI study in advance, using community meetings, indigenous leadership communications, and the available media. The urban

non-Indigenous population was recruited in the city of Juazeiro, Bahia. Neighborhoods in Juazeiro with a low migration profile were visited by the PAI staff and the habitants were invited to participate by community meetings and local political leadership communications. All invited individuals that met inclusion criteria and were willing to participate were included, after providing a written consent.”

- The groups assessed have different degrees of urbanisation; one group is defined as the ‘least urbanised group’, other as ‘intermediate degree of urbanisation’, and the other as ‘highly urbanised’. What is the proportion of urbanisation in each of those groups? This contextual information is important.

A: Thanks. We have adjusted the population groups section and explained the classification adopted.

- Body mass index is classified into 6 groups, but the sample is relatively small and some groups have a small number of participants; I suggest using only 3 categories (normal, overweight and obesity)

A: We agree with the reviewer and have adjusted this variable. We also corrected it in the results, in figure 3 and in table 1.

- Biomarkers were assessed in only a small part of the participants (varying from 6% in Truka men to 22.8% in Fulni-o women). It is unlikely that missing is at random, and those with information on biomarkers might differ from their original groups in several characteristics. Information regarding missing should be described.

A: We agree that this is a limitation of these data. We highlighted this limitation in the Discussion section, last paragraph: “There was a huge effort to obtain laboratory testing in remote Indigenous groups, but a high number of blood samples were later found inadequate for analysis due to difficulties in logistics and transportation. Therefore, as a substantial limitation, the number of laboratory results was inferior to the general clinical data. There are also intrinsic limitations related to a cross-sectional study, unable to address causality in the findings.

- Similarly, it seems that only 56% of the population had information on smoking, and this was only 24% in the non-Indigenous population. Why is there so much missing? Again, a flow chart would be helpful. The authors should reconsider the use of smoking and biomarkers information, given this is likely affected by selection bias.

A: The amount of unanswered smoking surveys can be explained by the fact that it is optional to answer the questions, and the subject can choose not to answer the item. We also agree that this is a limitation of the study. On the other hand, we believe that a total n of 558 respondents provides us with valuable information about smoking, especially since it identifies the high consumption of the traditional pipe (Xanduca) in the Fulni-O group, which has the best cardiovascular health status. This information is relevant for the development of future studies on the subject, since it opens a relevant research gap. We highlight this limitation in the discussion section.

Furthermore, we readjusted the illustrations: we reduced the content, merged figures 2 and 3 into one figure.

- Why non-parametric tests were used? Furthermore, several comparisons have been performed, which increases type 1 error. I suggest to use parametric tests and compare estimates with a single reference group (non-Indigenous).

A: We understand and agree with the reviewer's concern. However, in the study, the assumptions for applying parametric tests were violated: the distribution was different from a Gaussian curve and the variances across categories are not equal. To reduce type I error in multiple comparisons, we used a post-hoc test.

Since our goal is to compare different degrees of urbanization, and considering that Truká and Fulni-ô are in urban contexts, we compared the three groups.

- The results are confusing and seem to deviate from the objective of the manuscript, which was to describe the prevalence of cardiovascular risk factors in the different groups. The prevalence of several risk factors is also compared by sex within each group and sometimes between groups, and also by age groups (with pairwise comparisons). However, in addition to differing by degrees of urbanisation, the groups also differ in relation to sex and age. Therefore, age- and sex-standardised cardiovascular risk factors should be presented.

A: We appreciate the reviewer's concern about this issue. Indeed, it is very important, since cardiovascular diseases are influenced by age and sex, more stratified analyses according to these two variables were necessary. We have added a table with the rates of hypertension and diabetes mellitus according to sex and age group (table 3).

- The figures presented are not very informative. Results from figures 2-5 can be combined in a table.

A: The reviewer 2 suggested reducing the content of the illustrations, such as removing the values referring to the sum of the groups, which does not provide relevant information. In this sense, we adjusted the illustrations, merging figures 2 and 3 into one (figure 2). Figure 4 became figure 3, being also reduced in terms of content. Figure 5 became figure 4, also with reduced content. In this way, the text became uniform: three tables and three figures.

- "Cardiometabolic disease was observed in 40.4% of the total population". How was cardiometabolic disease defined?

A: Actually, the term "cardiometabolic diseases" refers to the prevalence of self-reported comorbidities (hypertension, diabetes mellitus, and/or dyslipidemia). We have corrected it in the text in order to avoid confusion of interpretation. Thanks for the comment.

- What is SAH (Figure 5)?

A: Systemic Arterial Hypertension. We have corrected it in the figure.

- It is surprising that 90% of the non-Indigenous population has high total cholesterol, and this prevalence seems to be <50% in the Brazilian population (PMID: 28658385). However, only 9.4% of the non-Indigenous population had information for total cholesterol. It is difficult to believe that these results (not only cholesterol, but also the other biomarkers) have not been affected by selection bias.

A: We agree with the reviewer that there is certainly an influence of selection bias and the number of individuals with information on laboratory tests. We have refined the discussion to make clear these limitations. On the other hand, we believe that it draws attention to a real existing problem, which is the effect of the urbanization process on cardiovascular health. From this, new studies are underway to better understand these issues.

Minor comments:

- The authors mention that Ingenious Fulni-o are the only Indigenous group who still use their original traditional language. Was the questionnaire applied in their original language?

A: The group, while preserving the traditional language, also communicates in Portuguese. We have added this information to the text.

- What is mean blood pressure (MBP)? Mean arterial pressure (MAP)? How was it calculated?

A: Thank you for your comment. It is actually the mean arterial pressure (MAP). We have added the formula used to calculate MAP in the methods section and corrected the term in the text.

Reviewer #2 (Remarks to the Author):

1. Abstract:

The proposed investigation of CV risk burden determinants was not object of the investigation, since no data were presented about mortality or disability. The abbreviation of kilo is kg, with lower case.

A: Thank you for your comment. The adjustments have been made in the abstract.

2. Methods:

2.1- The sample size needs more details. What was the total population of each group? What fraction of the population represents each study group? How was determined the sample size, in particular of the control group? How was the the refusal rate?

A: We appreciate the suggestions. We have added a paragraph explaining the selection process in Methods (Page 5): “All indigenous people living in the Fulni-ô an Truká communities were invited to participate in the PAI study in advance, using community meetings, indigenous leadership communications, and the available media. The urban non-Indigenous population was recruited in the city of Juazeiro, Bahia. Neighborhoods in Juazeiro with a low migration profile were visited by the PAI staff and the habitants were invited to participate by community meetings and local political leadership communications. All invited individuals that met inclusion criteria and were willing to participate were included, after providing a written consent.”

What was the reason to exclude those with clinically heart failure, coronary heart disease or cerebrovascular disease? If the purpose is a cohort study it seems reasonable to have population free of the disease at the begining, but not in a prevalence/cross-sectional study.

A: In this study, we looked for subclinical disease and risk factors. For this reason, individuals with these specific diagnoses already confirmed were not considered.

2.2- Laboratory testing. The normal values for HDL-c for non-fasting blood samples needs revision. The method for HbA1c needs to be informed since there is a great variability in the quality of this exam.

A: Thank you for your comment. This information has been included in the section on laboratory methods.

2.3- There is no information about the estimation of burden of CV risk factors.

A: We agree with the reviewer and removed the mention of burden of CV risk factors from the entire text.

3. Results

3.1- There is no reason to present average results for the sum of the three groups, since they are distinct groups.

A: Thanks for the suggestion. We removed from the text and figure 3. In addition, the figures were revised, being reduced. The old figures 2 and 3 were joined together and became figure 2.

3.2- Since the groups have distinct age-structure, prevalence rates should be presented as age-adjusted rates, or by age-groups.

A: Thanks for the suggestion. We have added this information in the Table 3.

3.3- p-values should be presented with p in lower case

A: Thank you for the comment. We have corrected it in the text and in the tables and figures.

3.4- When presenting prevalence rates of cardiometabolic diseases, it should be considered the cases that were excluded because they had these kind of diseases. The way was done induces some underestimation.

A: We agree with the reviewer. Actually, the term "cardiometabolic diseases" refers to the prevalence of self-reported comorbidities (hypertension, diabetes mellitus, and/or dyslipidemia). We have corrected it in the text in order to avoid confusion of interpretation.

3.5- No data for burden estimation, as mortality and disability, were presented

A: We agree with the reviewer. We have revised the text and removed everything that mentions burden estimation.

4. Discussion

4.1- In spite of sparse, there are several publications about hypertension in Brazilian Indigenous population after 2014.

A: Thanks. We updated the references, looking for the most recent ones.

4.2- Data presented did not allow to state that "CV risk burden is worse in Indigenous groups as the degree of urbanization is higher".

A: We agree with the reviewer. We have revised the text and removed everything that mentions burden estimation.

4.3- Among limitations of the study, it should be considered the sampling procedure used.

A: We have adjusted the text of the limitations and made the text clearer for the readers, including the limitations pointed out by the reviewers.

5. Figures and Tables

5.1- Figures are in excess and should not present results for the sum of the three groups.

A: We appreciate the suggestion. We have corrected and improved the quality of the figures.

5.2- Table 1 = categories of obesity should be grouped as obesity, since the number of individuals are fewer in some strata.

A: Thanks. We have corrected it.

6. References :

6.1- Need some up date

Thanks. We updated the references, looking for the most recent ones.

6.2- Reference 2 should have name off countries in capital letters.

A: Thanks. It was corrected

Reviewer #3 (Remarks to the Author):

Hi there,

I read your paper with interest. This paper looked at urbanisation and cardiovascular health among very interesting cohorts in Brazil. The paper is generally well written. The study design is appropriate for this research.

I have the following suggestions,

1. It is not clear how urbanisation is defined and measured in this study. A clear definition or discussion would be useful in this context.

A: Thanks for the comment. We have added more details about the included groups in the Methods section (Please see “Population Groups”).

2. To make it appealing to a wider audience, It would be interesting to discuss the findings here in relation to other similar studies looking at other developing countries, as similar findings were documented in other countries.

A: We appreciate the suggestion. The discussion section was updated.

3. Urbanisation may also bring positives that may benefit health outcomes, through new job, non-manual, opportunities and better infrastructure. So there might be a positive force associated with urbanisation on socioeconomic factors were neglected in the discussion. I would like to see some of these literature in the discussions.

A: Thank you for the comment. We have included a new statement in the discussion section.

Regards

Reviewer #4 (Remarks to the Author):

The current manuscript entitled “Urbanization and cardiovascular health among Indigenous groups in Brazil: The Project of Atherosclerosis among Indigenous populations (PAI)” aims to describe the prevalence of cardiovascular risk factors in groups of Brazilian Indigenous individuals at different stages of urbanization and investigate these groups’ CV risk burden. Although the topic is of great interest because it addresses Indigenous populations in Brazil, who are not contemplated in most epidemiological studies and whose epidemiological transition may be delayed according to urbanization, the burden of infectious disease, nutrition and access to health, it requires clarifications, and would also benefit from English editing. The authors concludes that the less urbanized Fulni-O ethnic group has a more favorable CV risk profile, when compared to the intermediately urbanized and the highly urbanized groups.

MAJOR

1. Although the findings are very interesting and corroborates the concept that urbanized lifestyles lead to loss of CV health – which is particularly relevant to indigenous individuals -, the effect of age should be better explored. It is expected that cardiometabolic risk factors increase with age and the age structure of the populations seem to differ. Fulni-O might be healthier just because they are younger, or it might be interesting to depict the later epidemiological transition if the younger have a worse profile than the older individuals, particularly if the age effect is different across populations. As such, I would like the authors to comment on two aspects:

1) for the comparison across groups, a better approach would be to adjust the findings regarding BMI, cholesterol, hypertension and diabetes for age;

A: We thank the reviewer for raising this very important point. We initially compared age within the three groups - the difference was observed between FULNI-O and control. However, the medians of age are close in both groups, which reduces the age bias. For the BMI variable, we present in figure 2 the comparison according to age group and in table 1 we compared according to ethnicity and sex. Unfortunately, the laboratory data have a limitation in the amount of data available and this prevents us from performing an adjustment through a multivariate model. However, we performed a better description of the data regarding these commented variables, and so we added a new table (table 3), which shows the distribution of SAH and DM according to age and sex.

2) It is interesting that younger individuals had higher BMI than their older counterparts. This could be a result of a more recent nutritional transition but may also be a selection bias or a survival effect bias, when the older individuals who were obese might have died. More detailed implications of the different populations' age structure should be added to the manuscript. Women might have this worse profile also because they were older than men.

A: This aspect is very relevant to the discussion of our findings. It is now more clear in the discussion section of this new version. Also, more details on the relation of age and BMI were provided in figure 2.

2. Although the authors argue that Fulni-O individuals had a better CV profile than the other comparison groups, still CV health is poor in the Fulni-O population, particularly when comparing to other data from cohorts and nationally representative surveys in Brazil. In the last VIGITEL study, the prevalence of self-reported obesity was 17%, while in the ELSA-Brasil cohort studies, which included individuals in 6 Brazilian capital cities, the prevalence of measured obesity was 22% (age range 35-74y, mean age 50y, 51% female). Even in the light of the known heterogeneity across the individuals included in these studies and the PAI study, it is important to highlight that the prevalence of obesity was considerably higher than the reported in other studies even for the Fulni-O population, despite a younger population. The same is true for abdominal obesity. This could be a result of the inclusion criteria, as such how the selection of the individuals for participation in the study was made should be further explained and evaluated as a possible limitation.

A: Thank you for your comment. We have discussed this issue further and have also expanded the limitations of the study.

3. Regarding BMI, two interesting points should be evaluated by the authors: 1) there were no individuals underweight, suggesting that the phase of malnutrition that usually occur in the first phase of epi transition has indeed been overcome; 2) Women had higher BMI than men: this is usually described in earlier stages of the second phase of nutritional transition – women go through the nutritional transition before men (maybe due to the fact that they have more sedentary roles). Another explanation may be due to selection

bias, as the findings differ from what was reported in the PNS, VIGITEL and ELSA-Brasil. Please elaborate on that.

A: We appreciate the pertinent comment. We have expanded the discussion and included these considerations.

3. Could the relation of hypertension with salt consumption be further explored?

A: Unfortunately, we don't have that information yet. However, there is an ongoing study on food consumption in these populations.

4. The losses for each analyses should be further detailed. It seems that only a minority of individuals had lab results, however this is not clear and it is a potential limitation that must be acknowledged.

A: Indeed, there were many laboratory data losses. We further detail this limitation of the study.

Reviewers' comments:

Reviewer #1 (Remarks to the Author):

In this revised version of the manuscript, the authors have addressed some of my comments and from the other reviewers. However, there are still several concerns regarding sampling, missing data and presentation/interpretation of the results.

From the description now added regarding the sampling process, it seems that the sample was a convenience sample, and therefore very likely to be biased. In the control group (non-Indigenous, urbanised population), neighbourhoods in Juazeiro with low migration profile were visited by the study staff and habitants were invited to participate. What does "low migration profile" mean and why only these areas were invited? How was this invitation? Are those who responded representative from the target population?

Selection bias has now been briefly mentioned as a limitation, but there is no attempt to discuss the direction of such bias.

There are important age and sex differences between the three groups assessed. Age- and sex-standardised cardiovascular risk factors should be presented to allow comparison across the groups, instead of stratified results.

Self-reported comorbidities were assessed, but blood pressure and glucose was also assessed in the study. Was comorbidity based on self-report only or also included measured blood pressure and glucose?

Why did the authors use self-report, given this is likely to be highly influenced by the degree of urbanisation (i.e. access to health care)?

The presentation of the results has changed, but it still does not represent the aim of the manuscript – to describe the prevalence of cardiovascular risk factors in three groups (two Indigenous and one non-Indigenous population) with differing levels of urbanisation. It would be helpful to have a table describing the characteristics of each population group (e.g., degree of urbanisation, age, sex, body mass index and other anthropometric parameters), and another table or figure presenting age- and sex-standardised cardiovascular risk factors, with comparison across the different groups.

It is very much appreciated the effort to collect blood samples in remote populations. However, a very high number of missing data is present for these parameters, and it is unlikely that this missing is at random. The authors should describe such missing to understand whether and how this influence the results.

The authors justified that smoking was an optional question (as were all the others, I imagine). However, the 44% who did not respond to the smoking-related questions (63% in the control population) is likely to differ from those who responded, and therefore estimates might be biased. The authors should consider whether these estimates are valid and discuss more about that.

Reviewer #2 (Remarks to the Author):

The revised form submitted presents improvement in the manuscript but still there are some problems as:

1. There is no information about the size and age/sex composition of the two indigenous groups. How was the procedure to obtain the sample of the non-indigenous group ? What is the explanation for the excess of women in the three populations ? How was the acceptance rate in each group ?
2. There is no information about procedures taken for the collect blood samples ? They were refrigerated or froen ? How long does it take from collection to analysis ?
3. Since there are a large amount of losses in blood sample results, the extrapolation for the whole sample has strong limitaions. The losses were higher for the man groups. For example, to estimate the prevalence of diabetes in the Truka man group the results of HbA1c is available only for 8 individuals and this is extrapolated for the 134 individuals in the group!

The size of the losses for laboratory data limit the extrapolation for the total sample.

I suggest a strong reformulation of the manuscript, exploring blood pressure (newly diagnosed hypertension, proportion on medication and under control, for example) and weight excess.

In spite of the importance of the study, the results need to be reliable.

I do not recommend the manuscript for publication in its present form.

Reviewer #3 (Remarks to the Author):

Thanks for this. Now I have no further comments.

Rebuttal Letter

Reviewer #1 (Remarks to the Author):

In this revised version of the manuscript, the authors have addressed some of my comments and from the other reviewers. However, there are still several concerns regarding sampling, missing data and presentation/interpretation of the results. From the description now added regarding the sampling process, it seems that the sample was a convenience sample, and therefore very likely to be biased. In the control group (non-Indigenous, urbanised population), neighbourhoods in Juazeiro with low migration profile were visited by the study staff and habitants were invited to participate. What does “low migration profile” mean and why only these areas were invited? How was this invitation? Are those who responded representative from the target population?

Answer – we thank the reviewer for his comments. In this new version, we clarify the recruiting procedures, by adding two last paragraphs in the ‘Population groups’ section (Page 6):

“When initial study planning was conducted, information regarding Juazeiro City districts was directly assessed in official public records available in the city hall. Recruiting was conducted in central neighborhoods of Juazeiro (Centro e Angari), with an estimate total of 3,270 inhabitants (similar to the 2570 inhabitants in the assessed Truka community and to the 3,254 Fulnio indigenous people). We selected two historical neighborhoods in the Juazeiro city, with lifelong residents. These areas are considered symbolic for the city growth, as most of the city expansion initiated from these central neighborhoods. Thus, these neighborhoods are the most representative of the original inhabitants of Juazeiro.

Importantly, in all three groups (Fulnio, Truka, and Juazeiro peoples), convenience sampling included men and women between 30 and 70 years old, to assess a more homogeneous cardiovascular risk base and to avoid age-related bias. Recruitment took place through invitations made to residents through neighborhood associations and local leaders, as well as radio and television networks. In the case of indigenous populations, prior contact was made with local leaders and visits were scheduled. None

of the people that voluntarily presented to the researchers refused to participate. All individuals who agreed to participate in the research provided a signed consent form.”

Selection bias has now been briefly mentioned as a limitation, but there is no attempt to discuss the direction of such bias.

Answer – we tried to further clarify this point, by adding more information in the limitation paragraph (Page 12): “Recruiting participants in indigenous settings is challenging, as it relies on voluntary participation and its potential bias. A higher number of female participants might have impacted in the cardiovascular risk burden, as men tend to have more risk factors and events in the studied age range.”

There are important age and sex differences between the three groups assessed. Age- and sex-standardised cardiovascular risk factors should be presented to allow comparison across the groups, instead of stratified results.

Answer – thanks for pointing out this important issue. We now added age- and sex-adjustments, as described in this revised Methods section and now shown in Figure 3.

Self-reported comorbidities were assessed, but blood pressure and glucose was also assessed in the study. Was comorbidity based on self-report only or also included measured blood pressure and glucose? Why did the authors use self-report, given this is likely to be highly influenced by the degree of urbanisation (i.e. access to health care)?

Answer – we thank the reviewer for the comment. In this version, we further clarify the variables description (Pages 6 and 7):

“Individuals were interviewed regarding diagnosis of hypertension and/or diabetes mellitus, as well as regarding the use of medications. Hypertension diagnosis was established if blood pressure equal or higher to 140x90 mmHg (28), as measured by the researchers or if the participant also referred the use of blood pressure medication. Mean arterial pressure (MAP) was obtained indirectly, through the following equation: $MAP = DBP + 1/3 (SBP - DBP)$, where: MAP = Mean Blood Pressure, SBP = Systolic

Blood Pressure and DBP = Diastolic Blood Pressure (27). Diabetes was established when HbA1c was equal to or higher than 6.5% or when the participant referred the use diabetes medications (26). Diabetes and hypertension prevalence were stratified by sex and age group (30-39 y/o; 40-49y/o; 50-59 y/o; and 60-70 y/o).

We also intended to investigate whether those participants that were aware of their CV risk factor would be adequately controlled. For this, those who auto-referred a known diagnosis of hypertension were assessed and classified as under control if BP < 140x90mmHg. ”

The presentation of the results has changed, but it still does not represent the aim of the manuscript – to describe the prevalence of cardiovascular risk factors in three groups (two Indigenous and one non-Indigenous population) with differing levels of urbanisation. It would be helpful to have a table describing the characteristics of each population group (e.g., degree of urbanisation, age, sex, body mass index and other anthropometric parameters), and another table or figure presenting age- and sex-standardised cardiovascular risk factors, with comparison across the different groups.

Answer – we now added a totally new Table 1 to address your concerns.

It is very much appreciated the effort to collect blood samples in remote populations. However, a very high number of missing data is present for these parameters, and it is unlikely that this missing is at random. The authors should describe such missing to understand whether and how this influences the results.

Answer - We agree that the laboratory missing data is indeed the most problematic part of our dataset. We agree that the high number of absent cases potentially jeopardizes the interpretation of the study results. Importantly, Reviewer 2 also draws our attention to this issue. For this reason, we decided to reformulate the text, excluding those variables that could jeopardize the study interpretation.

The authors justified that smoking was an optional question (as were all the others, I imagine). However, the 44% who did not respond to the smoking-related questions (63% in the control population) is likely to differ from those who responded, and therefore

estimates might be biased. The authors should consider whether these estimates are valid and discuss more about that.

Answer - We agree with the reviewer and revised the Discussion section in this new submission. This is, in fact, a complex issue to be analyzed. However, we believe that this information is valid, mainly because it demonstrates the high consumption of traditional pipes by the Fulni-ô population, and even so, this was the group with the best cardiovascular health profile. This may suggest that the mix herbs used may not be as harmful to health as commercial cigarettes. In addition, the lower prevalence of smoking in the city may reflect the effect of public anti-smoking policies, as shown in the recent literature regarding this subject.

Reviewer #2 (Remarks to the Author):

The revised form submitted presents improvement in the manuscript but still there are some problems as:

1. There is no information about the size and age/sex composition of the two indigenous groups. How was the procedure to obtain the sample of the non-indigenous group? What is the explanation for the excess of women in the three populations? How was the acceptance rate in each group?

Answer – we thank the reviewer for these important comments. In this new version, we clarify the recruiting procedures, by adding two last paragraphs in the ‘Population groups’ section (Page 6) and in the Discussion section (Page 10; last paragraph):

“When initial study planning was conducted, information regarding Juazeiro City districts was directly assessed in official public records available in the city hall. Recruiting was conducted in central neighborhoods of Juazeiro (Centro e Angari), with an estimate total of 3,270 inhabitants (similar to the 2570 inhabitants in the assessed Truka community and to the 3,254 Fulnio indigenous people). We selected two historical neighborhoods in the Juazeiro city, with lifelong residents. These areas are considered symbolic for the city growth, as most of the city expansion initiated from these central neighborhoods. Thus, these neighborhoods are the most representative of the original inhabitants of Juazeiro.

Importantly, in all three groups (Fulnio, Truka, and Juazeiro peoples), convenience sampling included men and women between 30 and 70 years old, to assess a more homogeneous cardiovascular risk base and to avoid age-related bias. Recruitment took place through invitations made to residents through neighborhood associations and local leaders, as well as radio and television networks. In the case of indigenous populations, prior contact was made with local leaders and visits were scheduled. None of the people that voluntarily presented to the researchers refused to participate. All individuals who agreed to participate in the research provided a signed consent form.”

“In addition, it is possible that the predominance of recruited women in all groups may be explained by the fact that men are usually less likely to pursue health care (7,10,12,15.”

2. There is no information about procedures taken for the collect blood samples? They were refrigerated or froen ? How long does it take from collection to analysis ? 3. Since there are a large amount of losses in blood sample results, the extrapolation for the whole sample has strong limitaions. The losses were higher for the man groups. For example, to estimate the prevalence of diabetes in the Truka man group the results of HbA1c is available only for 8 individuals and this is extrapolated for the 134 individuals in the group! The size of the losses for laboratory data limit the extrapolation for the total sample.

Answer - We agree that the laboratory missing data is indeed the most problematic part of our dataset. We agree that the high number of absent cases potentially jeopardizes the interpretation of the study results. Importantly, Reviewer 1 also draws our attention to this issue. For this reason, we decided to reformulate the text, excluding those variables that could jeopardize the study interpretation.

I suggest a strong reformulation of the manuscript, exploring blood pressure (newly diagnosed hypertension, proportion on medication and under control, for example) and weight excess.

Answer – we thank the reviewer for these suggestions. In this new submission, please find a totally revised version of our manuscript with additional information regarding the assessed risk factors.

Reviewers' comments:

Reviewer #1 (Remarks to the Author):

The authors have made considerable changes to the manuscript, which helped to improve its quality. The study is of great importance, especially considering the limited literature on Indigenous populations in Brazil. The sample was not representative (convenience sampling was used) and data was missing for several characteristics, some of which have been removed from the manuscript in this revised version.

However, I believe the manuscript would benefit from more changes, such as the ones detailed below:

- Blood markers were removed from this version of the manuscript due to the high amount of missing data. However, glycated haemoglobin was still used for the diagnosis of diabetes. It would also be important to revise the text, as those laboratory measures are still mentioned in the manuscript (e.g. limitations).
- The authors describe the prevalence of smoking, which also has a great amount of missing that is likely differential. Therefore, this prevalence is unlikely to be reliable. I suggest that the authors remove the smoking-related results and keep the manuscript limited to adiposity markers and blood pressure, for which data are available for most participants.
- Age- and sex-standardised prevalence of hypertension and diabetes are now presented (Figure 2). However, the standardisation has not been described in the methods. Which method was used to standardise that, direct or indirect standardisation? The authors should describe that.
- Table 1 has several p-values, which are confusing to interpret. I suggest that the authors only present a p-value for the comparison across groups, without multiple pairwise comparisons. Furthermore, some characteristics could be better explored, e.g., Ankle-brachial index (ABI) differs across the groups, but this difference is not observed as only one decimal place was used. For the continuous variables, the authors report both mean and SD median and interquartile range. It would be better to use the former for those normally distributed and the latter for asymmetrical distributions.
- Sorry for not noting this before, but why nonparametric tests were used instead of parametric? The sample is big enough and the distributions seem overall normal.
- On Table 2, I suggest having a p-value for comparisons across groups (as presented), and a p-value for sex differences. The authors report that the prevalence of obesity was higher in women than in men, but there is no formal comparison. On that note, why is there a comparison for BMI categories and a specific one for obesity? The authors should consider BMI as a single categorical variable.
- It is not clear why ABI differences across groups (Table 2) was not assessed. On the footnote it says that no test was applied, as it was within normality range. That does not justify not exploring differences across the groups. E.g., it seems that Fulni-o have lower mean ABI than non-indigenous in both males and females.
- I suggest that Figure 2, Figure 3 and Table 3 are removed (and their respective results). From Figure 3, the table now presented is very relevant, but it would be important to also have a comparison across groups (of age-and sex-standardised prevalence). As mentioned before, it would be important to consider not including diabetes.

With these suggested changes, the authors can focus more on the adiposity and blood pressure differences across the groups, and will be able to discuss in more detail about those differences and implications for health in these populations.

Reviewer #2 (Remarks to the Author):

The revised third version of the manuscript presents some answers to the comments from reviewers.

The sampling procedures, besides being non-probabilistic and having important excess of women, for the urban non-indigenous group has limitations to originate data for prevalence rates.

The three groups studied have different degrees of urbanization, which implies in different access to health care. The use of self-reported data about diagnosis of diseases may have different degrees of reliability and limiting comparisons among them.

Information about smoking for the entire groups are not accurate, since only 46.9% Fulni-ô, 29.3% Truká e 23.8% from non-indigenous group answered questions about smoking habits. This low and different rates of responses do not allow extrapolation for the whole group of participants. Data about smoking status presented in Figure 2, besides being with too much information are not friendly to comprehension, are presenting data with several restrictions in reliability.

Procedures for collecting, processing, storage and transportation of blood samples were not given. In this revised version, data on lipids were excluded. However, for diabetes are still presented in spite of HbA1c being performed from a small number of participants: for men, 15 Fulni-ô, 8 Truká and 2 non-indigenous; for women, 46 Fulni-ô, 19 Truká and 25 non-indigenous. These small number of exams, particularly for men, do not allow extrapolation results for the entire group. Interesting to note that HbA1c exams do not need fasting and the samples require only refrigeration, that is, there are no great difficulties to be used in field work and do not explain the low proportion of this exam in this survey.

REBUTTAL LETTER

Reviewers' comments:

Reviewer #1 (Remarks to the Author):

The authors have made considerable changes to the manuscript, which helped to improve its quality. The study is of great importance, especially considering the limited literature on Indigenous populations in Brazil. The sample was not representative (convenience sampling was used) and data was missing for several characteristics, some of which have been removed from the manuscript in this revised version. However, I believe the manuscript would benefit from more changes, such as the ones detailed below:

- Blood markers were removed from this version of the manuscript due to the high amount of missing data. However, glycated haemoglobin was still used for the diagnosis of diabetes. It would also be important to revise the text, as those laboratory measures are still mentioned in the manuscript (e.g. limitations).

ANSWER- Thanks for your comments. We excluded all information on laboratory testing.

- The authors describe the prevalence of smoking, which also has a great amount of missing that is likely differential. Therefore, this prevalence is unlikely to be reliable. I suggest that the authors remove the smoking-related results and keep the manuscript limited to adiposity markers and blood pressure, for which data are available for most participants.

ANSWER- Thanks for point out this important issue. We now revised the manuscript, complying with all recommendations.

- Age- and sex-standardised prevalence of hypertension and diabetes are now presented (Figure 2). However, the standardisation has not been described in the methods. Which method was used to standardise that, direct or indirect standardisation? The authors should describe that.

ANSWER- Very important point. We now clarified this information on the Statistics section.

- Table 1 has several p-values, which are confusing to interpret. I suggest that the authors only present a p-value for the comparison across groups, without multiple pairwise comparisons. Furthermore, some characteristics could be better explored, e.g., Ankle-brachial index (ABI) differs across the groups, but this difference is not observed as only one decimal place was used. For the continuous variables, the authors report both mean and SD median and interquartile range. It would be better to use the former for those normally distributed and the latter for asymmetrical distributions.

ANSWER- Thanks for the recommendations. We adjusted to comply with your comments.

- Sorry for not noting this before, but why nonparametric tests were used instead of

parametric? The sample is big enough and the distributions seem overall normal.

ANSWER- We tested the parameters and found asymmetrical distributions. Therefore, we performed nonparametric tests for group comparisons.

- On Table 2, I suggest having a p-value for comparisons across groups (as presented), and a p-value for sex differences. The authors report that the prevalence of obesity was higher in women than in men, but there is no formal comparison. On that note, why is there a comparison for BMI categories and a specific one for obesity? The authors should consider BMI as a single categorical variable.

ANSWER- Thanks for the recommendations. We adjusted to comply with your comments.

- It is not clear why ABI differences across groups (Table 2) was not assessed. On the footnote it says that no test was applied, as it was within normality range. That does not justify not exploring differences across the groups. E.g., it seems that Fulni-o have lower mean ABI than non-indigenous in both males and females.

ANSWER – we now added a test comparing the groups, but no statistical difference was found. It is now reported in the Results section.

- I suggest that Figure 2, Figure 3 and Table 3 are removed (and their respective results). From Figure 3, the table now presented is very relevant, but it would be important to also have a comparison across groups (of age-and sex-standardised prevalence). As mentioned before, it would be important to consider not including diabetes.

ANSWER- Thanks for your comments. We complied with your suggestions and excluded all information on diabetes.

With these suggested changes, the authors can focus more on the adiposity and blood pressure differences across the groups, and will be able to discuss in more detail about those differences and implications for health in these populations.

Reviewer #2 (Remarks to the Author):

The revised third version of the manuscript presents some answers to the comments from reviewers.

The sampling procedures, besides being non-probabilistic and having important excess of women, for the urban non-indigenous group has limitations to originate data for prevalence rates.

The three groups studied have different degrees of urbanization, which implies in different access to health care. The use of self-reported data about diagnosis of diseases may have different degrees of reliability and limiting comparisons among them.

ANSWER – thanks for your comments. We now used self-reported data exclusively on hypertension to expose the prevalence of indigenous peoples that are aware of this condition to investigate the proportion of controlled BP among them.

Information about smoking for the entire groups are not accurate, since only 46.9% Fulni-ô, 29.3% Truká e 23.8% from non-indigenous group answered questions about smoking habits. This low and different rates of responses do not allow extrapolation for the whole group of participants. Data about smoking status presented in Figure 2, besides being with too much information are not friendly to comprehension, are presenting data with several restrictions in reliability.

Procedures for collecting, processing, storage and transportation of blood samples were not given. In this revised version, data on lipids were excluded. However, for diabetes are still presented in spite of HbA1c being performed from a small number of participants: for men, 15 Fulni-ô, 8 Truká and 2 non-indigenous; for women, 46 Fulni-ô, 19 Truká and 25 non-indigenous. These small number of exams, particularly for men, do not allow extrapolation results for the entire group. Interesting to note that HbA1c exams do not need fasting and the samples require only refrigeration, that is, there are no great difficulties to be used in field work and do not explain the low proportion of this exam in this survey.

ANSWER- Thanks for your comments. We excluded all information on diabetes and smoking.